# Friction Behavior of Rough Surfaces on the Basis of Contact Mechanics: A Review and Prospects

**DOI:** 10.3390/mi13111907

**Published:** 2022-11-04

**Authors:** Siyuan Zhang, Dawei Li, Yanwei Liu

**Affiliations:** 1School of Mechanical Engineering, University of Shanghai for Science and Technology, Shanghai 200093, China; 2Department of Mechanics and Engineering Science, College of Engineering, BIC-ESAT, Peking University, Beijing 100871, China

**Keywords:** rough surface, material plasticity, contact, friction, interface slip

## Abstract

Contact and friction are closely related as friction cannot happen without contact. They are widely used in mechanical engineering, traffic, and other fields. The real contact surface is not completely smooth, but it is made up of a series of tiny contact asperities as viewed in the micro-scale. This is just the complexity of the contact and friction behaviors of rough surfaces: the overall mechanical behavior is the result of all asperities which are involved during the contact. Due to the multiplicity of surface topography, the complexity of contact scale, and the nonlinearity of the constitutive material, there are still many open topics in the research of contact and friction behavior of rough surfaces. Based on the perspective of the macroscopic and micro-nano scale contact mechanics, this review gives a brief overview of friction for the latest developments and points out the existing issues and opportunities for future studies.

## 1. Introduction

Contact and friction are closely related to the engineering industry, especially in the technologies associated with rolling bearing, wheel–rail control, head–disk interaction, etc. Therefore, which factors are related to the contact mechanical properties of solid surface and how to design and control the interface friction coefficient are hot issues in scientific research and engineering practice. In fact, the contact behavior and friction properties are closely related to the microstructure of the contact surface. Based on the accurate characterizations of the surface topography, the corresponding research can start from the local contact model, investigating the contact and friction relationships of each contact asperity of the rough surface, and finally accurately predicting the contact relationships and friction performances of the interfaces. 

A typical application of contact and friction is wheel-rail contact [1]. The research of wheel-rail contact stress, contact state and friction coefficient, plays a very important role in the improvement of train running speed. In the fields of bolt connection [2], magnetic head reading and writing [3], and artificial joints [4], the influence of friction needs to be eliminated. The key to the process is to reduce the coefficient of friction. Hydraulic seals and elevators and other instruments [5] need to use friction, and the key to its process is to improve the coefficient of friction. In addition, friction occurs during the gear transmission process. By using the friction force between the ship and the pile, the ship is wound on the pile. The friction piles of the bridge use friction to support vertical loads. According to statistics, one-fifth of the world's annual energy consumption is used to overcome friction in systems and processes [6]. Therefore, it is of great significance to reduce and control friction in order to save energy, prolong the life of machinery and equipment, and protect the environment. 

Friction cannot occur without interfacial contact. The difference between them is that the former has a tangential force and the latter has only a normal force. Contact mechanics is the foundation of tribology. Therefore, the discussion of friction studies often relies on contact mechanics. Whether analyzing the contact state or the friction performance of the contact asperities, the contact and friction phenomenon has been investigated for more than one hundred years, and these research results are still playing an extremely important role in life and engineering. In this field, the research trend has the characteristics of ‘from macro to micro, from qualitative to quantitative, from single mechanical discipline to highly inter-discipline’. The research field of contact mechanics and tribology is expanding from the analysis of contact and friction phenomena to the combination of analysis and control. In addition, the research in this field has also developed rapidly, and many complex contact and friction behaviors can be quantitatively simulated and calculated with high precision, in order to provide more effective guidance for scientific research and engineering practice. 

In this paper, the research progress of contact and friction behaviors of rough surfaces is summarized from three aspects: contact responses of a single contact asperity, the contact between rough surfaces, and friction behaviors of interfaces. The recent advances are reviewed and some research prospects are presented in each section, and finally, we summarize our views.

## 2. Research Progress of Contact Modeling for Single Asperity

The real contact surface is never completely flat. From a microscopic point of view, a real rough surface can be seen as a series of tiny asperities that are uneven. By establishing the mechanical model of a single asperity, the results can be applied to more complex statistical models, multi-scale contact models, and so on [7]. Thus, the mechanical properties of real rough surface can be investigated in depth. Since the rough surface has the nature of an uneven interface, the roughness often leads to a very small contact area, resulting in relatively high local contact pressure and contact stress, leading to interface failure or yield. In many cases, especially metal surfaces, failure or yield often occurs in the contact surface region, and plastic deformation often occurs as well. Therefore, it is necessary to take purely elastic contact, elasto-plastic contact, and completely plastic contact into consideration in the study of contact properties of single asperities. 

Generally speaking, the Hertz solution is used as the reference solution for the elastic contact of the slightly convex body, and the solution methods for contact asperities with different shapes are also based on the extended Hertz solution. However, it is difficult to apply the analytical solution to general cases because it is usually suitable for the case at small load. Since the analytical solutions of the contact problems [8] are limited to special geometric profiles and constitutive behaviors, most contact investigations take plasticity into consideration by combining finite element methods or elastic theory with approximations (often referred to as semi-analytical models) [9,10]. For finite element simulations, it is necessary to use appropriate geometry, boundary conditions, and appropriate material properties. The parametric study is often carried out by changing the material properties and geometric parameters. Then, the contact force, deformation, and contact area obtained by simulations are analyzed and processed dimensionless. These dimensionless treatments are often associated with known elastic solutions (such as the Hertzian contact theory) or with the onset of plastic deformation (such as the appearance of a yield point), which can be analyzed with a specific yield criterion (such as Von Mises yield criterion) and an appropriate elastic solution. For fully plastic contact, the results and trends are usually analyzed by the finite element method (FEM), and then obtaining the empirical equations. The main purpose of presenting empirical equations is to predict elastic-plastic contact behavior in scientific and engineering applications, consider the interactions between rough surfaces, and control plastic deformation, friction, fatigue, and wear between solid surfaces through rough surface interactions. In this section, the research progress of single and multi-asperities contact models is reviewed according to the geometry of contact profiles.

### 2.1. Two-Dimensional Contact Model

In actual contact conditions, the real rough surface is usually highly anisotropic, thus a two-dimensional contact model cannot be used in rough surfaces. The model of cylindrical contact asperities in contact with a rigid flat is not as common as that of three-dimensional contact models. In fact, the two-dimensional contact model is also suitable for some special cases, such as wheel contact, interaction between gear teeth, cylindrical roller bearings, and so on. 

Cylindrical contact, namely, the two-dimensional plane strain contact problem, is also called the line contact model, or two-dimensional plane strain contact model. Figure 1 shows a schematic of cylindrical contact, which is also known as the plane strain contact problem. Many researchers believe that most actual contact between cylinders is plane strain (when both ends of the cylinder are rigidly fixed) [7,8], thus there is no strain outside the section perpendicular to the symmetry axis of the cylinder. The elastic solutions of two-dimensional plane strain contacts (cylindrical contacts) are well known and are often classified as Hertzian solutions. 

In addition to theoretical research, there are also some works which have studied the numerical solutions of the two-dimensional plane strain contact problem by means of finite element analysis. The study by Dumas et al. [11] is one of the early important studies on cylinder contact. In addition to the indentation problem between a rigid cylinder and a deformable body, it is also necessary to consider the contact problem between a deformable cylinder and a rigid plane, which is a more common and widely used simplified model. Later, Cinar and Sinclai [12] also conducted similar research. Cinar et al. [12] extended the maximum load and also adopted the Von Mises yield criterion and the incompressibility of deformable materials, taking the effect of strain hardening into account. In fact, this conclusion is not reasonable: with the improvement of simulation accuracy, the average contact pressure is not a fixed value. Under fully plastic contact conditions, the average contact pressure will decrease with the increase of the contact area and the loading [13]. Later, Sharma et al. [14] investigated the contact behavior of elastic-perfectly plastic (1% bilinear hardening) materials under plane stress through finite element analysis. They considered a wide range of material properties and proposed corresponding empirical formulas. The average normal pressure (hardness) of cylindrical indentation contact in plane stress predicted by finite element method is much smaller than that of the spherical indenter. 

Most of the studies mentioned above predict the relationships between contact area, contact force, and loading displacement of plane strain models at macroscopic scale by using theoretical analysis and finite element methods. They often use Von Mises yield criterion to determine when plastic deformation occurs, and use these critical values to make the finite element results dimensionless, thus it can be used more conveniently in engineering fields. However, with the development of research, these empirical formulas at macro-scale cannot meet the needs of engineering devices, as the device size has the tendency of miniaturization. Obviously, the reduction of device size will lead to size effect, and the contact relationship predicted by the traditional constitutive method will gradually fail. Although the two-dimensional contact problem considering the couple stress (a mechanical theory of elasticity at micro and nano scale) is not a new problem, the derivation process is extremely complicated and the computational simulation is difficult to reproduce, resulting in it being difficult to popularize the corresponding solutions. Therefore, there are still lots of unsolved problems in two-dimensional contact mechanics at micro and nano scales, especially the computational simulation methods and experimental designs are still challenging.

### 2.2. Three-Dimensional Contact Model for Spherical Shape

From the above literature review, from the microscopic scale, rough surfaces can often be considered as a series of random micro-asperities. Therefore, the contact relationship of individual asperity is a very important problem in contact mechanics and has been the focus of researchers for more than a century. People theoretical analysis, numerical simulations, and experiments to analyze and predict various contact properties, such as real contact radii, stress distributions, and contact forces. 

In order to investigate the contact relationship of individual asperity, some assumptions and simplifications should be made about the geometry of the single contact asperity. The most common simplification is to assume that the individual contact asperity has a spherical profile, as shown in Figure 2. The spherical asperity model has been used by many researchers from different fields, such as tribology [15], electrical contact [16], etc. Although spherical contact asperities are often used for contact studies on rough surfaces, they can also be used to consider particle flattening and interactions between surfaces, for example, anisotropic conductive films [17], wear particles [18], and nanoparticles [19].

Focusing on different simplification methods and contact conditions, Johnson summarized the early research progress of contact mechanics in detail [8], and introduced the conventional methods for solving two-dimensional contact problems and three-dimensional contact problems. In addition to the theoretical solution of pure elasticity, a large number of works have studied the elastoplastic contact, plastic contact, and other problems. Sinclair et al. [20,21] first proposed a model for an elasto-plastic body in contact with a rigid sphere, and divided the contact model into flattening model, indentation model, and more complex model. Jackson and Kogut [22] compared the flattening and indentation problems of a single spherical contact asperity. They revealed the different contact behaviors with respect to the constitutive relations of contact parameters and deformation states quantitatively, and compared them with empirical models based on the finite element method. In our study, we compared the contact characteristics of the flattening model and the indentation model by using the finite element simulation method, as shown in Figure 3. The results show that when the hemispheres or the substrate are elastic, they can be equivalent when loading displacement is small. When taking plasticity or complicated loading conditions into consideration, the contact behaviors of them are quite different.

With the development of finite element technology and the improvement of calculation accuracy, the flattening model and indentation model of a single spherical asperity emerged successively. Chang et al. [23] proposed the concept of elastic-plastic contact by taking the elasto-plastic sphere in contact with a rigid flat surface as an example, but the first derivative of contact load and contact area was discontinuous. Zhao et al. [24] adopted a mathematical processing method to express contact load and contact area transition smoothly. In the study of the asperity flattening model, the most classic research is the finite element model proposed by Kogut and Etsion [25] (KE model for short). The KE model revealed three different stages—from purely elastic to elastic-plastic and then to fully plastic contact interface. The empirical formulas of contact load, contact area, and mean contact pressure versus dimensionless penetration depth are provided, and the classical Hertzian solution is extended to fully plastic contact. Compared with the previous models, the KE model is much more accurate, and the relatively accurate contact state boundary and boundary basis are proposed for the first time. Chatterjee et al. [26] investigated the effect of different tangent moduli on the contact properties of elastic-plastic spheres considering strain hardening. However, for the indentation model, it is important to determine the relationship between the mean pressure and the yield strength: Jackson et al. [9] studied the indentation problem of a rigid ball using the slip line theory. If the flattening model and the indentation model are regarded as two extreme models in the study of contact models, Gheadnia and Jackson et al. [27] proposed an elasto-plastic contact model between them. It is worth noting that the entire deformation occurs on the sphere for the flattening model, which is different from the indentation model. Therefore, these two models cannot be regarded as completely equivalent models. In addition, the flattening model itself is finitely large, while the indentation model itself is a semi-infinite space. The former cannot release the plastic deformation, and the influence of the constraint at the bottom of the hemisphere on the result cannot be ignored, while the latter has enough space to release the plastic deformation, and the semi-infinite space will also have subsidence. In addition, during the deformation process, the curvature of the flattening model changes, while the indentation model stays the same. Since the flattening and the indentation model of a single spherical contact asperity are different to some extent, the contact conditions and relative strength of the surface should be considered when selecting the flattening model or indentation model. 

On the other hand, some studies have investigated the contact relationships of a single spherical contact asperity in the framework of the couple stress theory. Zisis et al. [28] derived the general solutions of several basic contact problems under the framework of the couple stress theory and described the relevant size effects in the microstructure. Karuriya et al. [29] solved the contact problem of a finite thickness layer and indenter with different shapes in the same framework. Zisis et al. [30] investigated to what extent gradient theory could capture indentation responses and related size effects in microstructure materials in the experiments. Peng et al. [31] studied the size effect in spherical contact using the modified couple stress theory.

Regardless of the rationality of their basic assumptions and the accuracy of their results, the above studies all emphasize the importance of spherical contact under different simplification methods and loading conditions. Empirical formulas for predicting contact relationships are often given, but there are still some problems. In fact, the classical Hertzian solution can only be applied to small deformations, and the current research lacks a clear discussion on the applicable scope of the Hertzian solution. A great deal of deformation often occurs when the yield point appears, especially when the plasticity index of the material is small. There is no accurate prediction to describe the contact relationship in the existing research work. In addition, due to the fact that the size effect cannot be ignored at micro and nano scales and the limitation of analytical and computational methods, there are still many open topics on spherical contact analysis based on size effect theories, such as gradient elasticity theory and gradient plasticity theory, which is now and then still a research hotpot.

### 2.3. Three-Dimensional Contact Model for Sinusoidal Profile

In addition to the spherical profile assumption mentioned above, Bush et al. [32] extended the generalized spherical contact asperity to a paraboloid with the same principal curvature and applied the classical Hertz solution to its deformation (referred to as the BGT model). Other researchers have simplified the geometry of a single contact asperity into ellipsoid [33,34], parabola [35], cone [36], sinusoidal [37], and so on.

Compared with the spherical asperity hypothesis being used to describe the rough surface, using sinusoidal profile is better. On the one hand, it can be seen from the rough surface contours measurement methods that the sinusoidal shape is more realistic than the spherical description of a single contact. For example, Poon et al [38] measured the profile of rough surface by using an atomic force microscope (AFM for short). In our study, we use white light interferometer to measure metallic surface (Figure 4a) and we generated numerical rough surface to simulate the interfacial mechanical behaviors (Figure 4b), which is similar with Yastrebov et al [39]. These experimental and simulation measurements show that using sinusoidal profile to describe a single asperity is better. On the other hand, from a computational point of view, sinusoidal contact asperities are better than spherical contact asperities: sharp angles at the connection between the hemisphere and the substrate will cause stress concentration when plasticity is considered, whereas sinusoidal profiles have no such problem because their geometry is continuous. In fact, sinusoidal descriptions of asperities (Figure 5) have been used in many studies of the mechanical responses of individual contact asperities [40,41,42,43].

The finite element and boundary element methods are often used in this part of research. For example, Krithivansan et al. [44] used the finite element model to study the mechanical response of elastic-plastic sinusoidal contact pairs. At initial contact, the contact behavior of sinusoidal profile is similar to the spherical contact, and both of them can be described by the classical elastic Hertzian solution. Once the pressure exceeds a certain deformation range, they behave very differently. This model gives an approximate relation between contact area and contact pressure and an empirical expression of the average pressure leading to perfect contact. Gao et al. [42] analyzed the two-dimensional contact between rigid plane and elastic-plastic sinusoidal surface, and studied the parameters of plastic deformation and residual stress. The contact types of two-dimensional sinusoidal surface models are defined. For each contact type, the relationship between contact pressure, contact size, effective indentation depth, and residual stress is discussed in detail, which provides a research basis for the sinusoidal contact research. William [45] established a plastic model to describe the pressure and contact area of a local sinusoidal surface. It is proved that the assumption of perfect plastic behavior of rigid bodies tends towards infinite pressure when the surfaces are close to full contact (full contact is defined as the ratio of contact area to nominal contact area close to 1). Liu et al. [46,47] solved the contact problem between a rigid flat surface and a sinusoidal contact asperity by using finite element simulation. The initial critical plastic variables, including loading displacement, contact radius, first yield depth, and pressure, were obtained by fitting the relationship curves between the simulation results and the material and geometric parameters. Saha et al. [40] analyzed the contact problem between a rigid flat surface and a three-dimensional sinusoidal contact asperity through finite element simulation, and gave empirical expressions for the average pressure in the full contact state.

From the above literature review, on the one hand, it is more realistic to describe a single contact asperity by sinusoidal profile than by spherical shape. On the other hand, for elastic contact, almost all studies considering sinusoidal contact asperities take the classical Hertz solution as the reference solution, whose basic idea is that the tip of sinusoidal asperity can be regarded as the tip of a sphere. What is less clear from the current study, however, is when the deformed contact asperities violate the assumption of a spherical shape. This indicates that an analytical solution based on Hertzian solution is needed to describe the elastic contact response of a single sinusoidal asperity. Based on this, the analytical solutions of sinusoidal asperity can be extended, especially when taking adhesion, roughness, and even friction into consideration. It is worth noting that most of the existing studies on asperities with sinusoidal shape are carried out at the macroscopic scale. It is necessary to accurately describe the contact behavior of individual sinusoidal asperities at different scales in order to carry out the contact map of rough surface at micro and nano scales.

## 3. Study on Contact between Rough Surfaces

In this section, we introduce the research of rough surface contact and ways that single asperity models are included in rough surface contact frameworks. Since the real contact surface has varying degrees of roughness. Thus, the real contact area, is often much smaller than what it appears to be. It is very difficult to consider roughness in contact models due to the complexity of randomness and features over multiple scales. Many researchers have focused on this problem, and there are many models to predict the contact relationships between rough surfaces.

### 3.1. Surface Roughness

One of the fundamental problems in contact modeling between rough surfaces is the realistic representation of roughness. The real contact area, as well as the distribution and morphology of the real contact spots, affect many contact mechanical mechanisms. Surface roughness affects the stress state and interface stiffness of the contact interface. In order to understand the effect of roughness on contact interface information, tribology and wear, it is necessary to accurately describe the surface topography accurately. 

In the roughness characterization problem, amplitude ($S_{a},\;S_{q},\;S_{sk},\;S_{ku}$), spatial ($S_{al},\;S_{tr},\;S_{td}$) and hybrid parameters ($S_{dq},\;S_{dr}$), or Abbott–Firestone (bearing area) curve-based parameters ($S_{k},\;S_{pk},\;S_{vk}$, material ratios, and volume parameters for 3D measurements), demonstrate the complexity of reaching a universal description of surface roughness [48,49]. Indeed, in most contact models, only a small subset of these parameters is used. 

In numerical simulations, fractal parameters are often used to characterize the surface roughness. With the introduction of fractal roughness, numerical models began to use PSD to fully define surface roughness. However, the PSD does not represent complete information about the surface topography: the same PSD can generate different surfaces in real space [50]. There are also many studies on the experimental measurement of roughness. The Abbott and Fells Universal pen recorder measures surface roughness by drawing an enlarged version of the "stylus" (broken blade) motion on the surface [51]. Then, a multitude of techniques were developed to measure roughness: contact and optical profilometry, stripe projection scanning, scanning probe microscopy (SPM), transmission electron microscopy (TEM), etc. [52,53]. These techniques, both contact and non-contact, have some limitations that should be carefully considered when interpreting data. For example, reference [54] for white light interferometry and [55,56] for scanning force microscopy. 

No matter if the surface roughness is measured experimentally or simplified by numerical simulation, the characterization of surface topography is a very important problem. Therefore, the following two sections review the research progress of contact behavior of rough surfaces from two aspects: statistical modeling and fractal modeling.

### 3.2. Rough Surface Research Based on Statistical Model

In the field of contact problems on rough surfaces, an important issue is the distribution and evolution of contact spots. The most classical rough surface model is the statistical model [57] proposed by Greenwood and Williamson in 1966 (GW model for short). Many studies focused on rough surfaces are based on the GW model, and more general results are often obtained by releasing the basic assumptions of this statistical model. Although the surface topography of rough surfaces can be measured by atomic force microscopy, it is still important to establish reasonable numerical models of rough surfaces. For the numerical modeling of rough surfaces, one of the most important problems is to characterize the roughness of the contact interface properly.

The schematic of the GW modeling method is shown in Figure 6. This model is based on the following basic assumptions: (i) the tiny asperities are regarded as hemispheres of the same size, whose height follows Gaussian or exponential distribution; (ii) the deformation of the whole rough surface is assumed to be small and the mechanical response of a single asperity can be described by classical Hertzian solutions; and (iii) elastic interactions between adjacent asperities are not considered. The GW model predicts the linear relationship between the actual contact area and the normal contact force. However, the assumptions of GW models are very strong, thus it is unrealistic, this is why some researchers have tried to release these assumptions to carry out more universal models.

In order to describe the surface topography of rough surfaces based on statistical models, Feng et al. [58] used N, β, and σ to describe the roughness of the contact interface, where N, β, and σ are relevant to the surface spectrum. Some researchers use the ratio of the standard deviation of the height distribution of asperities to the mean curvature of the asperities $\sigma /\bar{R}$ to characterize the surface roughness [59,60,61,62]. The maximum value of $\sigma /\bar{R}$ is 0.5. Since the most common simplification is to assume that the rough surface is composed of a series of spherical asperities, which have the same radius and their heights are randomly distributed. Therefore, $\sigma /\bar{R}$ is often used to characterize the roughness of the surface [63,64], and the magnitude of $\sigma /\bar{R}$ is usually less than $10^{-2}$.

After proposing reasonable roughness characterizing parameters, the geometric characteristics of a random rough surface can be obtained, thus the contact characteristics of a rough surface can be investigated further. The complexity of the rough surface contact problem is that, the overall behavior of the surface is the composite result of all asperities involved in the contact process. Therefore, it is necessary to describe the geometric properties and mechanical behaviors of individual asperities accurately, and it is important to describe the interaction between asperities during the deformation process. 

The profile assumption of single asperity is essentially the problem of rough surface characterization, while the small deformation assumption is essentially the problem of loading condition. In addition to releasing the first two assumptions, the interaction between asperities can also be considered. Zhao et al. [65] simulated the influence of interaction on local deformation according to Saint-Venant principle and Love formula. Ciavarella et al. [66] calculated the deformation when considering the interaction by distributing the contact pressure over the contact area evenly. Song et al. [67] used statistical methods to find the displacement of neighboring asperities during contact, thus effectively adjusting the average surface spacing. Ciavarella et al. [68] proposed a discrete GW model, considering the interaction effect with the calculation method of loading the displacement field outside the circle, and compared the results with the fractal surface. Yeo et al. [69] proposed a discrete model considering interaction, and used the solution of substrate deformation to adjust the normal position of each contact point in the process of quasi-static contact, thus obtaining the interaction force on the surface. Chandrasekar et al. [70] established a finite element model to verify the discrete model considering the interaction. Li et al. [71] used the discrete model method to consider the elastic interaction between asperities, and discussed the influence of the number of asperities considering interaction. 

In addition, the traditional GW model only considers the elastic contact, while the plastic deformation will inevitably occur in the real contact situation. When taking plasticity into consideration, Kogut and Etsion [25] proposed the fitting formula of contact force and contact area based on finite element simulations, and then applied the fitting formula in the statistical GW model to predict the surface contact response. Similar to KE, Song et al. [72,73] studied the effect of size-dependent plasticity on the contact behavior of rough surfaces. Zhang et al. [74] analyzed the deformation process of rough surfaces by using the strain gradient plasticity constitutive theory (CMSGP for short) by using a finite element method. They developed a full-scale elastoplastic contact model at different size levels.

As previously stated, for a statistical-based model, it is more realistic to describe the geometry of individual contact asperity in sinusoidal shapes, and current methods for characterizing the roughness of surfaces do not adequately characterize rough surfaces composed of sinusoidal asperities. As a result, it is necessary not only to propose an appropriate roughness characterization method for rough surfaces composed of sinusoidal asperities, but also to concentrate on the corresponding contact relations when the interface roughness is high. As the mechanical response of individual asperity is dependent on the model's geometry and boundary conditions, it is obvious that the model will fail when the deformation exceeds a certain range (i.e., the assumption of an equivalent radius of curvature fails). When considering plasticity and size effect, even if asperities of different shapes can have the same equivalent radius of curvature, the mechanical responses are totally different. As a result, the contact behaviors of the rough surface should be investigated in detail after characterizing the surface morphology properly. 

Furthermore, regardless of the accuracy of the assumptions and results, attempts to study the interaction effects based on the rough surface models mentioned above emphasize the importance of the interaction between contact asperities. However, the models mentioned above have some issues in common. The interaction effects between asperities are averaged over the surface, which is the only choice of statistical models. The interaction effect is primarily reflected in contact delay. This interaction effect, however, is physically localized and diminishes as the contact fraction (the ratio of contact pairs where contact occurs to total contact pairs) increases. Interaction not only causes a delay in contact, but it also alters the height distribution of contact asperities and, as a result, the distribution of local contact points/areas. New models that can obtain information about each contact asperity must be proposed in order to capture the local characteristics of the interaction effect and changes in the asperity height distribution. The effect of the interaction, together with nonlinearity of material and size effect, should be investigated in depth in the future. 

### 3.3. Fractal Modeling Method for Rough Surface

Despite rough surface contact models derived from the GW model [75,76,77], some researchers are committed to the study of multi-scale model and fractal model, etc., and different research methods have triggered a series of related discussions [78,79,80,81]. Fractal theory was proposed by Benoit B. Mandelbrot in the late 1970s. It has been applied in many scientific fields and solved a large number of difficult problems. Many researchers applied fractal theory to the field of tribology, which made people pay attention to the great value of fractal theory in the study of contact characteristics of microscopic rough surfaces [82,83,84,85]. A fractal surface has the following characteristics: (1) it has a precise and meticulous composition, and the details can still be observed after several magnifications. (2) It is irregular and has complex geometric features in both the overall outline and local details. (3) It has self-similarity, and the structure of things is similar under different magnifications. (4) In general, it can be composed by continuous generation. The schematic diagram of the fractal surface profile is shown in Figure 7. 

The contours of many engineering surfaces have self-affining fractal characteristics, thus many researchers have studied the contact behaviors of fractal surfaces. For example, the Persson contact theory [86,87] does not use the basic assumptions based on the GW model, and believes that the roughness of the interface gradually increases with the increase of contact load. Of course, this assumption is reasonable under full contact conditions, but the prediction of the contact relationship under small loads is not accurate enough. Kogut et al. [88] compared the differences between statistical models and fractal models, and believed that the difference in contact relationship was only related to the difference in the characterization methods of contact surfaces. In addition, Majumdar and Bhushan [89] constructed the MB model by using the W-M function, which is used to characterize the two-dimensional contour topography of isotropic fractal rough surfaces, and combining with the fractal theory to extend the mechanical characteristics of a single asperity to the entire contact surface. Yan et al. [90] extended the two-dimensional profile to the three-dimensional contour surface, described the actual rough surface more accurately, and modeled the contact mechanics of the contact surface. Wang et al. [91] took the influence of interaction into account and modeled the normal contact stiffness. They found that the normal stiffness of the contact surface would be overestimated when ignoring the interaction effect. Yuan et al. [92] and Xu et al. [93] divided the asperities into different grades according to the cosine wave scale of the rough surface contour, carried out the fractal modeling of contact force and contact stiffness of the contact surface, and studied the influence of the grade range on the contact characteristics of the contact surface. Chung et al. [94] proposed a fractal model of contact mechanics in which the micro contact asperities of the contact surface were elliptic, but this model considered that the eccentricity of all elliptical contact points was equal, which was obviously too idealistic. Chen et al. [95] proposed a fractal contact model considering detailed multi-scale effects, and they analyzed the influences of topography and material properties. Yu et al. [96] proposed a novel multi-stage contact model of fractal rough surfaces. They gave actual asperity deformation and actual deformation mode depending on contact degree. They established a complete model of contact characteristics across multi-stages. 

Although there have been many studies on rough surface contact modeling, the modeling method based on statistical model inspiration and the modeling method based on fractal theory are still two parallel modeling methods, as shown in Figure 8. For the study of the interface’s mechanical behavior at micro and nano scales, the importance of multiscale roughness has long been recognized. However, the accurate description of rough surfaces at different scales is still active and there are many unsolved open topics. This is helpful when facing the next frontier of the contact modeling challenge, which is to describe the contact of rough surfaces during sliding and thus study the frictional behavior between rough surfaces.

## 4. Research Progress on Interface Sliding and Friction Behavior

The classical friction law states that the friction force is proportional to the normal pressure, that the friction force is independent of the contact area, and that the friction coefficient is constant [97]. As research progresses, many new findings show that the coefficient of friction is affected by the interface material, normal pressure, and sliding velocity [98,99]; the coefficient of friction is not an inherent property of the material [100] but a function of the whole system. 

Since one-quarter of the global energy loss is caused by friction and wear, it is important to reduce frictional wear and lower the coefficient of friction [101]. Lu Ke's group [102] proposed methods to prepare new materials that significantly reduce the coefficient of friction, the use of plasticity accumulation caused by rough contact during sliding is important. Friction cannot occur without sliding, thus the cause of sliding is also important. It has been suggested that sliding is caused by the spalling of the leading edge of the slip nucleus driving the slip nucleus behind it forward [103].

In engineering practice, the use and prevention of sliding friction are common, but the understanding of the sliding friction coefficient has taken several twists and turns; in the design of engineering systems, it is crucial to further understand the interface friction and sliding law from the contact study of single contact asperity.

### 4.1. Friction and Slip Behaviors of a Single Asperity

Interfacial slip is the result of a combination of factors, including material properties, interface roughness, local slip, chemical reactions, etc. The force required for sliding has been a hot topic in the past decades. The classical study of asperity sliding originated from the Mindlin model in the 1940s [104]. According to the Mindlin model, the contact region between two spheres consists of a central adhesion zone surrounded by an annular slip band. As the tangential load increases, the central adhesion region gradually decreases and eventually disappears because the material within the central adhesion region cannot withstand infinite traction. In contrast to Mindlin's analytical solution, Hamilton derived an explicit expression for the stress field under non-axisymmetric sliding contact [105]. 

Chang et al.’s [23] CEB friction model is one of the most famous friction models for asperity contact, as shown in Figure 9. They assumed that sliding happens at the moment of yielding and they used the von Mises yield criterion and Hamilton's stress field to determine the maximum tangential load for a single contact asperity. Since the first yield point of a spherical contact is surrounded by a large elastic region, their results underestimate the maximum tangential load that a single contact asperity can carry. 

When taking plasticity into consideration, Kogut et al. [106] combined finite element analysis with analytical expressions to provide a semi-analytical solution for the sliding inception of an elastoplastic spherical contact. In another study [107], Brizmer et al. considered local slip effects and extended the study to elastoplastic spherical contacts. They concluded that local slip conditions have a greater effect on the evolution of the plastic zone as the normal load increases. Other models have tried to discuss whether the fully adherent case [108] or the fully slippery case [109,110] could be applied to real interfaces. In other studies, slip initiation conditions between deformable spheres (cylinders) and rigid flat plates were studied using finite element methods. Brizmer studied sliding friction under high adhesion conditions [111] and extensively investigated the effect of plasticity on friction parameters such as junctional tangential stiffness, static friction force, and static friction coefficient; full adhesion conditions were used, assuming high adhesion strength. In a similar study [112], Wu et al. investigated the adhesive friction of an elastoplastic cylinder in contact with a rigid flat plate, establishing a link between static friction and ductility. Later, in order to better characterize the friction process, Shi and Wu et al. [113,114,115] relaxed the all-sticky model to a partial slip state (the finite element model is shown in Figure 10), and their proposed model revealed the friction transition between the KE model [106] and the all-sticky model proposed by Brizmer et al. [111] (BKE model for short). Xie et al. [116] carried out single-asperity friction simulations by using a molecular dynamic simulation method. They analyzed stress distribution and the dislocation evolution process of single asperities. However, from the above literature review, the competing mechanisms between material plasticity and the effect of partial slip on slip initiation have not been analyzed in depth. Further studies should be carried out to investigate the friction mechanisms.

### 4.2. Examples of Interface Friction Behavior Standing on Contact Mechanics

Based on the friction behavior of a single asperity and the contact mechanics of rough surfaces, Xu et al. [117] proposed a statistical model of the contact area and friction of a soft rough interface in shear, describing the interplay between adhesion and friction at the microcontact scale. The macroscopic friction response is explained by microscopic experimental phenomena combined with microscopic scale parameters and surface statistical data. Maier et al. [118] determined the contact pressure of real rough surfaces. They coupled the micro-model and the macro-model to consider the influence of different geometry scales on tribology behaviors. Jamshidi et al. [119] developed a hybrid contact model of a hard rough surface on an elastic substrate by using the multi-asperity contact theory. They obtained interface physical characteristics such as initial tangential interface stiffness, slippage friction force, and maximum pre-slip displacement based on the asperity contact theory. Wu et al. [120] presented a statistical analysis for the microscale flattening of random surface asperities. They developed a multiscale soft-contact modelling method for rough surfaces contact, in order to characterize the rough surfaces in contact with coupled slipping/sliding and rolling. Salari et al. [121] introduced the research progress of frictional creep model based on the rough elastoplastic contact model. Kang et al. [122] established a new contact model for rough interfaces, which was characterized by multiscale topography parameters. They revealed the effects of the multiscale topographies on the normal and tangential contact responses. Mergel et al. [123] captured shear-induced contact area reduction in Hertzian rubber contact, and they incorporated adhesion, friction, and large deformations. They developed a computational framework, which is suitable to investigate sliding friction even under normal tensile loads. Hu et al. [124] discussed the friction behavior between an elastic body and a rigid body with a random rough surface based on contact mechanics. According to the characteristics of random rough surface, they carried out numerical experiments on several settings, such as ‘rough–smooth’, ‘smooth–rough’, and ‘rough–rough’ contact, and discussed the affecting factors of the global coefficient of friction. These are typical examples of friction models of rough surfaces based on contact mechanics, which provides many theories and methods for further study of surface friction behavior. 

### 4.3. Experimental Methods for Friction Studies

In addition to theoretical studies and numerical simulations, the evolution of interfacial friction can also be revealed from an experimental point of view. In order to fully understand the tribological phenomenon and deeply explore the mechanism of friction, wear, and lubrication, it is necessary to carry out relevant tests and analyses. The development of modern surface testing technology and instruments provides an effective means to observe surface phenomena and their changes at the atomic and molecular scales, which makes it possible to carry out experimental studies on tribology at different scales. Examples of the main instruments used for tribology experimental research are as follows: friction and wear testing machine, fretting friction and wear testing system, scanning probe microscope, friction force microscope, etc.

Typical experimental studies on friction coefficients are fretting friction experiments. Fretting is caused by the periodic relative displacement between two objects that are in contact. When the reciprocal motion has a sufficient number of cycles, wear occurs on or near the contact surface. Before wear occurs on the material surface, the fretting process depends on a number of factors dominated by the normal load, the tangential displacement amplitude, and the friction coefficient. Although it is possible to model the contact interface under such loading conditions, it is difficult to specify the value of the friction coefficient in the model (which may vary in space and time) without the help of experimental studies. Therefore, it is important to carry out fretting experiments. The fretting experimental setup which is commonly used is shown in Figure 11. 

In terms of material selection for fretting experiments, the most commonly used materials are titanium alloys [125,126], followed by nickel alloys [127,128], pure copper [129], polymers [130], etc. Commonly used titanium alloys (Ti-6Al-4V) have low density and good mechanical properties and are commonly used in the dovetail contact area of gas turbine blade/disc attachments. However, titanium alloys have poor tribological properties and tend to have premature crack sprouting due to fretting. 

As for fretting experimental methods and measurement equipment, traditional measurement methods, such as extensometers [131] and differential transformers [132], can be used to investigate fretting tribological properties by measuring tangential contact stiffness, but these measurements usually underestimate the stiffness values because of the inclusion of volumetric flexibility contributions. In addition to traditional methods, techniques such as laser interferometry [133] and ultrasound have also been used to investigate the tangential stiffness of contact surfaces [134,135,136,137]. 

Oxford University has a long history of studying fretting fatigue [138]. Kartal et al. [127] developed a digital image correlation (DIC) technique to characterize the frictional properties of rough contact surfaces, using optical methods to overcome this limitation of not being able to measure full-field displacements. 

Mulvihill et al. [139] compared the frictional properties obtained by DIC and ultrasonic contact stiffness measurements obtained by ultrasonic techniques. Through finite element simulations and experiments, Mulvihill et al. [140] concluded that friction is caused by plasticity and tangential interface adhesion. Eriten et al. [141] investigated the effect of surface roughness on energy dissipation in friction problems between rough surfaces, and verified the analytical result that rough surfaces dissipate more energy than smooth surfaces. Mulvihill et al. [128] investigated the increase in frictional forces during the fretting process as a result of wear-scar interactions.

Cao et al. [129] investigated the surface morphology and the microstructure evolution of pure copper materials. They investigated the friction mechanism in the wear phase and its influence on the subsequent steady state. Jin et al. [142] compared experimental and finite element simulation results to investigate the effect of partial slip on the performance of fretting and established a fretting diagram by using fretting fatigue life, relative slip range, and normal force. Through simulations and experiments, Yue et al. [143] concluded that the use of variable friction coefficient has little effect on the amount of wear at the end of the steady-state phase of the fretting.

Since the contact interface in engineering is not completely smooth, sliding and friction between two rough surfaces are important [144,145]. In sliding contact problems, where sometimes friction needs to be reduced [146], and sometimes friction needs to be exploited [147], the traditional Cullen's law of friction is not applicable in all cases [148]. Despite the long history of tribological research and the great progress made in recent years in the contact mechanics of rough surfaces, there are still many open topics in the field of friction on rough surfaces [149]. The friction analysis based on statistical models is still the most commonly used in current friction studies on rough surfaces [150,151,152,153,154].

In fact, the physical processes that determine friction characteristics are complex: there are contact interactions [70], adsorption, adhesion [155], material properties, etc. It is difficult to predict the friction coefficients of actual friction pairs by means of simplified numerical models and to really understand what the main control parameters are. Therefore, it is necessary to take the simplified friction model commonly used in two rough surface sliding problems as an example to reveal the influencing factors of interfacial sliding and friction coefficients from theoretical, simulation, and experimental perspectives by releasing the assumptions of traditional friction models for material and interfacial sliding criteria. 

Most current theoretical and simulation studies on interfacial sliding and friction mechanisms investigate the plastic effect in a fully adhesion model or local slip in a Mindlin framework. There are few studies on the mechanism of the transition from plastic yielding to full slip at the early stage of sliding, and an accurate prediction of it is still lacking. Therefore, models are needed to analyze the competing mechanisms between these two factors and provide a quantitative friction diagram; and investigate the slip transition process from complete slip to full plastic yielding by comparing the evolutionary relationship between plasticity and slip. For the common simplification of the sliding friction model for two rough surfaces, the current study lacks the analysis of the whole sliding process for two hemispheres in the case of lateral contact, as well as the theoretical analysis of hemispheres and planes in the state of complete sliding, and the related predictions need to be supported by friction experiments, and the corresponding friction phenomena need to be analyzed by theoretical models. 

## 5. Prospects of Research Themes in Contact and Friction Behaviors of Rough Surfaces

Starting from a single contact asperity, the present review focuses on the study of surface contact and friction behavior, which provides ideas and methods for the study of rough surface contact and friction. From the above literature review, the contact and friction behaviors of rough surface were summarized. It is an effective way to realize the reasonable prediction of surface contact and friction behaviors by proper analysis and simulation methods. It is also the basis of understanding interface friction behaviors at micro and nano scales and describing complicated multi-physical phenomena at different scales. However, the results of contact or friction behavior predicted according to the above different modeling methods are very different, which increases the difficulty for researchers to choose a reasonable modeling method. Due to the complexity of surface topography, the diversity of contact scales, and the nonlinearity of material constitutive relationships, the difficulty of establishing accurate prediction models is significantly increased, as shown in Figure 12. 

In the future, more complex and in-depth research should be carried out in many aspects in the future, since the research trend has the characteristics: from static to dynamic, from qualitative to quantitative, and from macro to micro. Therefore, the future research direction can be explained in a general way. 

(1) The research of contact mechanics and friction behaviors of rough surfaces has a trend from macro to micro. The actual contact area of the real rough surface is the sum of the areas of several microcontact asperities, thus the relatively high microcontact asperities will be the first to enter the plasticity. In the micrometer scale plastic behavior, the contact plastic behavior will involve the size effect. The role of size-dependent parameters in the contact and friction properties of rough surfaces is a focus of future research. 

(2) The research of friction behaviors of rough surfaces has a trend from qualitative to quantitative. Starting from the classical friction law, researchers not only reveal the causes and influencing factors of friction qualitatively, but also devote themselves to putting forward prediction formulas and criteria that can be directly used in engineering practice. The quantitative prediction of interfacial friction is of great significance for extending the study of contact mechanics and friction mechanics to engineering practice.

(3) The research of friction behaviors of rough surfaces has a trend from a single mechanical discipline to highly inter-disciplined. Tribology is integrated with chemistry, biology, environmental science, MEMS, and other disciplines. New disciplines such as tribology, biological tribology, and ecological tribology have been formed. 

With the rapid development of intelligent science and technology, device miniaturization and new materials continue to emerge, which puts forward higher requirements for the research of surface mechanics. The study of contact and friction between solid surfaces has a trend from macroscopic scale to micro and nanometer scale. The traditional continuum mechanics theory cannot explain the mechanical problems that may be encountered in the future. Since classical continuum mechanics is unable to characterize the mechanical behavior at micro and nano scales, the research on the surface properties of materials represented by contact mechanics and frictional properties is in urgent need of follow-up. The contact and friction phenomena at different scales are of great significance. Thus far, however, the present studies focus on analyzing contact behaviors at the framework of strain gradient theory, or revealing the micro mechanism of friction at the atomic scale level. They did not reveal size effect in the process of friction mechanism and the origins of the static friction. The different scales of friction behavior research still remain to be further developed and improved. Since the reduction of system size and contact size, the problems related to friction response at micro-scale have become more prominent, which puts forward a higher requirement to deeply understand the physical nature of the multi-scale friction behaviors and mechanisms of both traditional and new materials, such as micro and nano structure materials, biomimetic materials, gradient nanostructured materials, and so on. Based on the proper modeling of material interfaces and the mechanics theory at different scales, the friction behavior and the theoretical framework across scales should be established. The understanding of the friction origin at different scales should be deepened, and more accurate and practical friction prediction methods should be explored. It is very important to guide the evaluation of the tribological properties of traditional and advanced materials. It is also of critical importance to promote the application of tribological theory in the reliability assessment of micro-and nano-scale devices.

## 6. Conclusions

In recent years, many scholars have carried out a lot of research on the contact and friction behaviors of rough surfaces. This review briefly summarizes the current progress in this field from three aspects: contact responses of a single contact asperity, contact between rough surfaces, and friction behavior of interfaces. Presently, a lot of research is still needed to explain the influence of plasticity, size effect, and surface roughness on interface friction behavior and to reveal the basic processes and internal mechanisms of contact and sliding in the process of shearing. It is of great significance in theoretical research and engineering application to systematically understand the interaction mechanism, effect of roughness, slip, and plastic size effect in the friction process. It is also a challenge faced by future tribological research.

## Figures and Tables

**Figure 1 micromachines-13-01907-f001:**
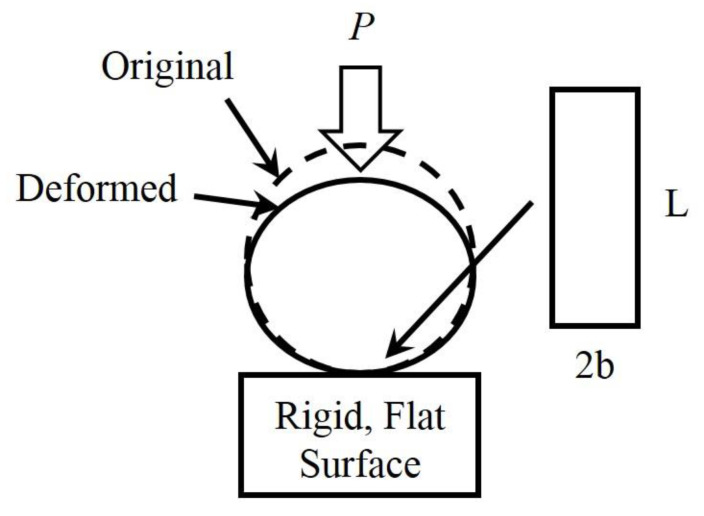
Schematic of contact between a cylinder and a rigid flat surface.

**Figure 2 micromachines-13-01907-f002:**
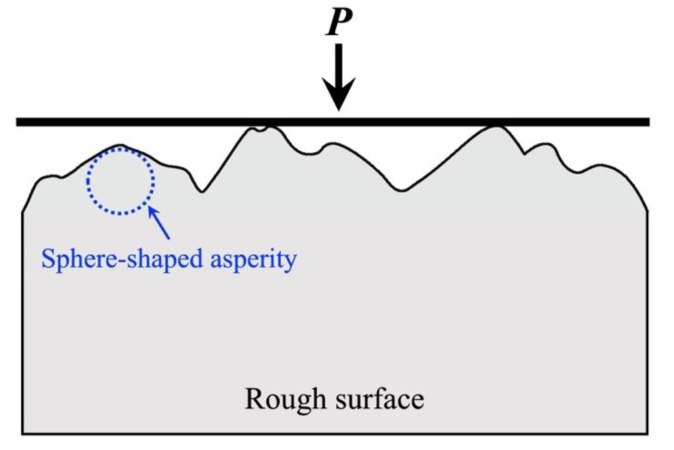
Contact schematic of multi-asperities rough surface: a real rough surface is in contact with a rigid smooth, which can be seen as made up of a series of spherical asperities.

**Figure 3 micromachines-13-01907-f003:**
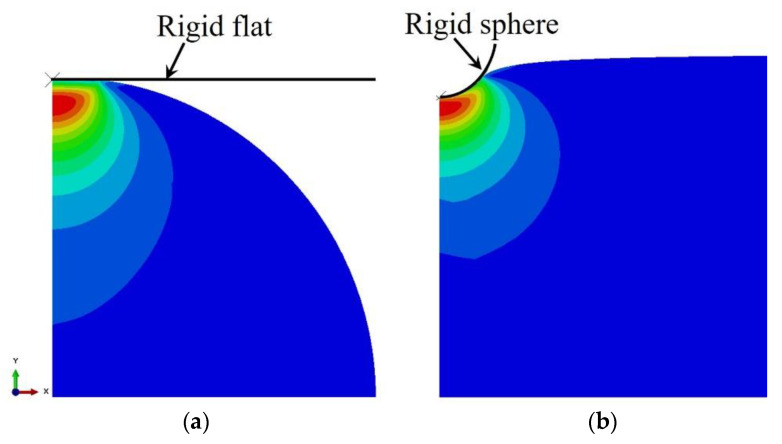
Finite element simulations of (**a**) a deformable asperity (flattening model) and (**b**) a rigid indenter (indentation model).

**Figure 4 micromachines-13-01907-f004:**
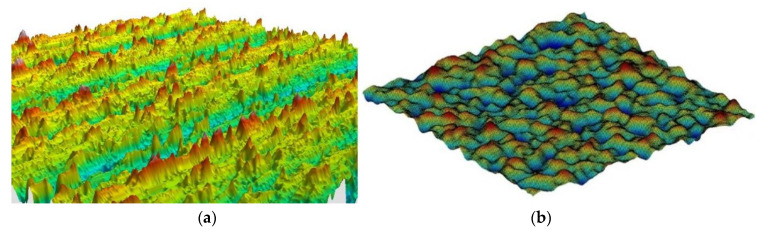
(**a**) Surface profile of metal, which is scanned by white light interferometer; (**b**) examples of rough surfaces generated by computers.

**Figure 5 micromachines-13-01907-f005:**
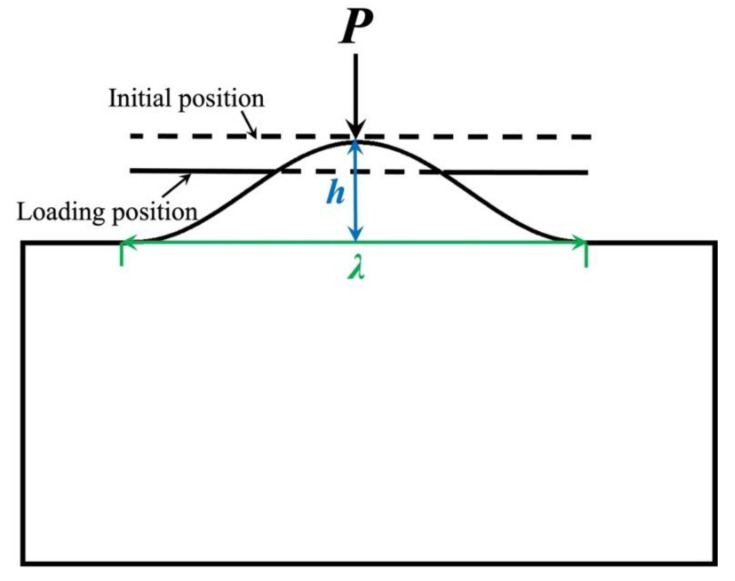
Schematic of a sinusoidal asperity loaded with a rigid flat surface.

**Figure 6 micromachines-13-01907-f006:**
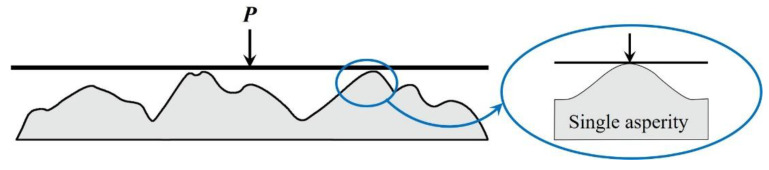
Schematic of GW modeling of rough surfaces and a single contact asperity.

**Figure 7 micromachines-13-01907-f007:**
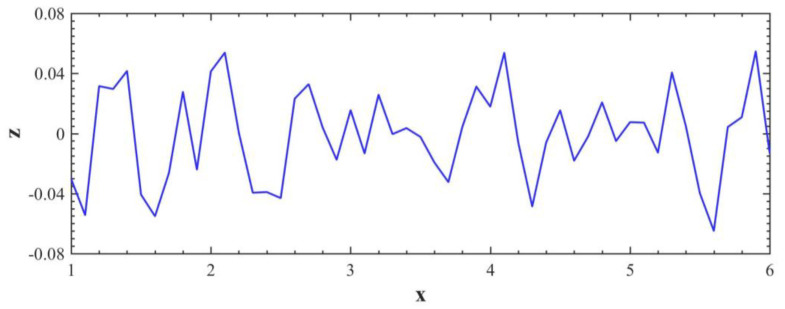
Schematic of fractal surface profile.

**Figure 8 micromachines-13-01907-f008:**
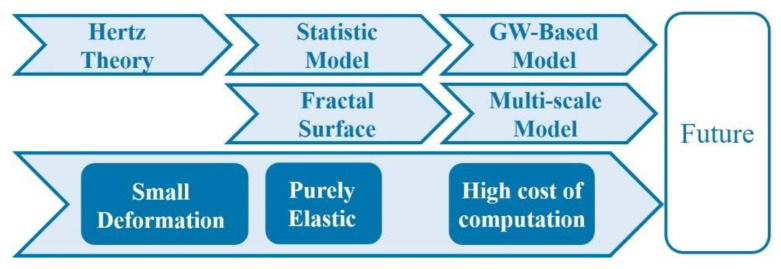
Two parallel modeling methods and limitations.

**Figure 9 micromachines-13-01907-f009:**
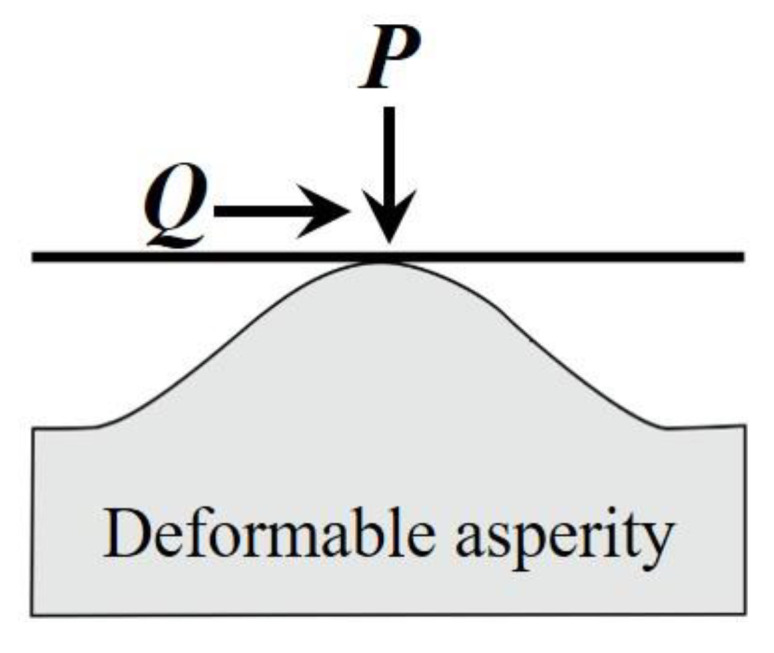
A deformable asperity in contact with a rigid flat surface under combined normal and tangential loading.

**Figure 10 micromachines-13-01907-f010:**
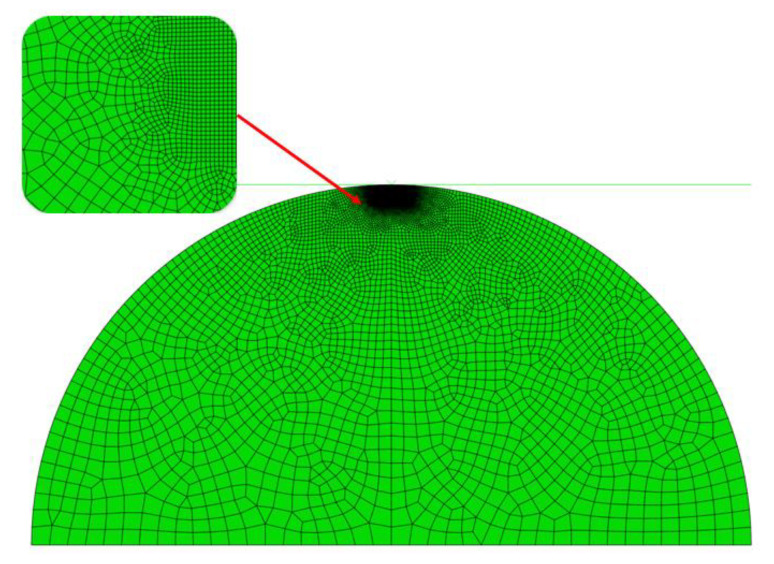
Schematic of finite element modeling of a spherical asperity.

**Figure 11 micromachines-13-01907-f011:**
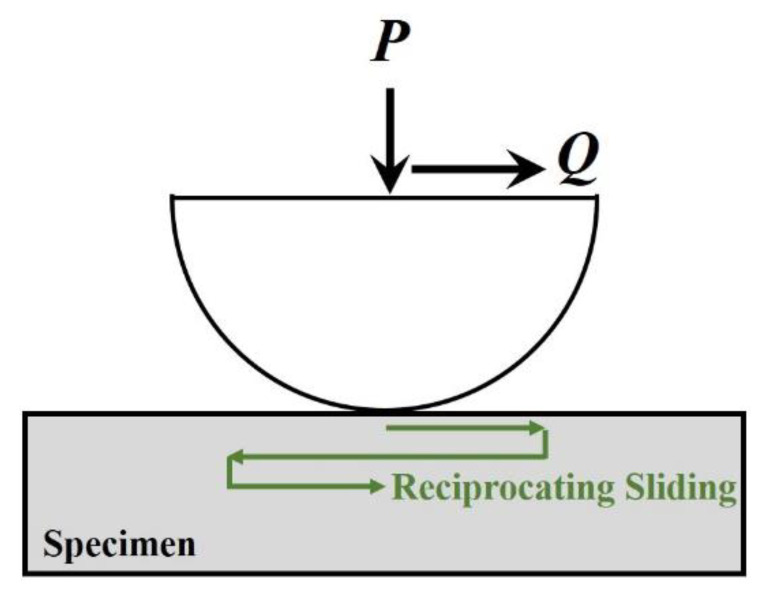
Schematic diagram of fretting experiment.

**Figure 12 micromachines-13-01907-f012:**
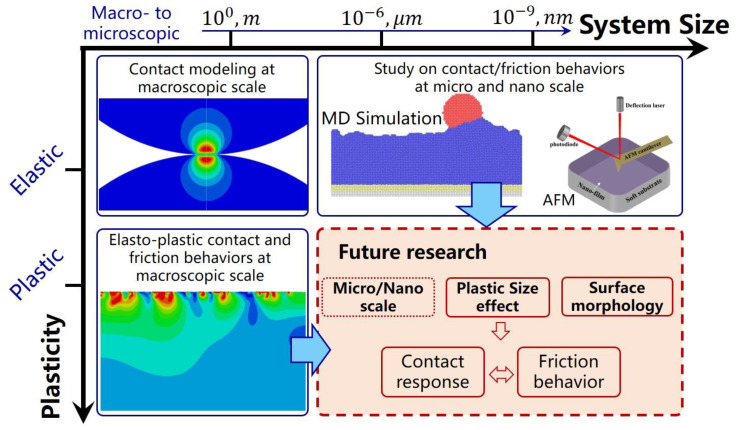
A plasticity vs. length-scales map of models developed in tribology: elastic contact modeling at macroscopic scale, elasto-plastic contact and friction behaviors at macroscopic scale, simulation [156] and experimental methods at micro and nano scale, and future research focus.

## Data Availability

Not applicable.

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
