# Peer review of "Friction Behavior of Rough Surfaces on the Basis of Contact Mechanics: A Review and Prospects"

_micromachines, 2022, doi:10.3390/mi13111907_

Round 1

Reviewer 1 Report

The purpose of this paper is not clear. There is no clear link and no clear differentiation between the two topics "contact" and "friction behavior" – it’s more about the consideration of real surface topography within these fields - “contact” and “friction”. Furthermore, "nano scale" is only mentioned rarely but is part of the headline. Many mistakes with references and unclear connections between figures and text are problematic. "Recent advances" are mentioned in the title but only less than 25% of the cited references are of the last five years. In general, the title is a little misleading. For a “general” review paper experimental methods should not only include fretting tests – there is a lot of different experiments concerning friction behavior (gears, clutches, brakes, tires, bearings, sealings, …). Measurement of rough surfaces is partly mixed up with the simulation (see for example in figure 13) – this is not a study on elastic contact or friction. Abbreviations are not explained at first appearance in text. Often examples are very uncommon or interesting parts are not explained well. Captions for figures are not formatted correctly/consistent.

In this stage, the recommendation is to majorly revise structure, content, and purpose of this paper. The purpose of the paper should be stated clearly in the abstract and introduction. Attached, you can find a selection of annotations that should help to understand the above-mentioned reasons for a major revision.

Author Response

We thank the reviewer for your constructive and detailed comments.

Answer to the comment:

(1) The occurrence of friction is inseparable from interfacial contact. The research on friction and sliding is usually based on the research on interface contact. These two aspects have a very close connection and distinction. This has been emphasized in the introduction and the main text. The revised text has been highlighted in blue.

(2) The friction research at nano scale is often carried out by molecular simulations. The relevant research progress has been mentioned, but does not emphasize the scale range.This point has been emphasized and explained again in the revised manuscript, and the revisions have been shown in blue text.

(3) As for literature, we have added some references in the past five years and added some new arguments. The text of the modified part has been shown in blue text.

(4) There are indeed many experimental methods in tribology field. In this paper, only one of them is selected to be elaborated in detail, and the rest of the experimental and simulation methods are outlined in Section 4, which have been highlighted in blue.

(5) The abbreviations and their related explanations and illustrations have been checked and modified one by one.

(6) After reading the whole text, we revised the text again to emphasize the main content and writing purpose of this paper. 

Other explanations can be seen in the attachment and the manuscript. 

We thank the reviewer again for your detailed suggestions. 

Reviewer 2 Report

1.For the depth of the current research is not enough, it is only a brief introduction, and no constructive research direction is given.

2.For different individual asperity shapes, not just spherical vs sin shapes. Where are the disadvantages of parabolic shapes, elliptical shapes, and other shapes?

3.For the contact of the rough surface, it is not a simple contact, the influence of the interaction of the asperities on the contact, and the influence of the lubrication conditions on the contact need to be analyzed in depth?

4.The author describes the future research direction in a very general way. Can you give three specific research directions?

Author Response

Point 1: For the depth of the current research is not enough, it is only a brief introduction, and no constructive research direction is given.

Response 1: We thank the reviewer for this comment. We have proposed some future research focus at the end of each section and presented our views in Chapter 5. In order to make it more explicitly, we have added some descriptions and elaborate the future research hotspots and developing directions in Chapter 5. The revisions have been marked in blue text.

Point 2: For different individual asperity shapes, not just spherical vs sin shapes. Where are the disadvantages of parabolic shapes, elliptical shapes, and other shapes?

Response 2: We thank the reviewer for this comment. For elasto-plastic contact of a single asperity, the mechanical response is clearly different for different shapes of asperities even though they have the same radius of curvature at the asperity tip. Therefore, it is inappropriate to represent the asperity by a sphere. Moreover, the problem of using spherical shapes, parabolic shapes, elliptical shapes and other shapes in FEM simulation is the sharp corner between the asperity base and the substrate: these sharp corners create stress concentrations which yield plasticity during the asperity deformation with very small load, as shown in Figure 1. The convenient way of avoiding this problem is by using a geometry which has C1 continuity, such as sinusoidal profile.

Figure 1. Stress concentration of spherical model causes plasticity at the corner, only a small portion of the substrate is shown here. (See attachment)

Point 3: For the contact of the rough surface, it is not a simple contact, the influence of the interaction of the asperities on the contact, and the influence of the lubrication conditions on the contact need to be analyzed in depth?

Response 3: We thank the reviewer for this comment.

From a microscopic point of view, a real rough surface can be regarded as composed of tiny contact asperities with random distribution of geometric properties (such as height, size, etc.), which play an important role in contact-related engineering. Therefore, it is very important to understand the mechanical properties of rough surfaces composed of these tiny contact asperities. However, the complexity of rough surface contact problem lies in that the overall contact behavior of the surface is the result of all contact asperities involved in the contact process. Therefore, a reasonable prediction of contact behavior requires not only an accurate description of the geometric properties and mechanical behavior of individual contact pairs (as stated in Chapter 2), but also an accurate description of the interaction between these tiny contact asperities during deformation.

Among rough surface contact models, one of the most famous theories is Greenwood and Williamson's statistical method (GW model for short). The theory treats tiny contact asperities as hemispheres of the same size, whose height follows a Gaussian or exponential distribution. Furthermore, it is assumed that the overall deformation is sufficiently small and that no elastic interaction occurs between adjacent contact asperities. The GW model successfully predicted the linear relationship between the actual contact area and the contact force. However, the assumptions of statistical method are too strong and in most cases very unrealistic, which is why some researchers have tried to release them. Attempts to study interaction effects based on statistical models, regardless of the accuracy of their assumptions and results, emphasize the importance of interactions between contact pairs. However, the above models share some common problems :(1) the interaction effect between contact pairs is averaged through the surface, which is the only choice for statistical models. (2) The interaction effect between contact pairs in the statistical model is mainly reflected in the contact delay. However, this interaction effect is highly localized in physics and diminishes as the contact fraction (the proportion of contact pairs in which contact occurs to the total number of contact pairs) increases. The interaction is not only the delay of the contact, but also changes the height distribution of the contact pairs and thus the distribution of the local contact points/regions. In order to capture the local characteristics of contact pair interaction and the variation of contact pair height distribution, a new model that can obtain the information of each contact pair needs to be proposed. Therefore, it is important to consider in detail and accurately calculate the effect of interactions on rough contact. In addition, the influence of lubrication conditions on contact is also very important, but it is not within the scope of this paper, so it will not be elaborated on lubrication

We have added some explanations in Section 3 in the revised manuscript. The revisions have been highlighted in blue.

Point 4: The author describes the future research direction in a very general way. Can you give three specific research directions?

Response 4: We thank the reviewer for this comment.

Starting from a single contact asperity, the present review focuses on the study of surface contact and friction behavior, which provides ideas and methods for the study of rough surface contact and friction. However, more complex and in-depth research can be carried out in many aspects in the future. Since the research trend has the characteristics: from static to dynamic, from qualitative to quantitative, from macro to micro. Therefore, the future research direction can be explained in a general way.

(1) The study of interface contact and friction has a trend from macro to micro. The actual contact area of the real rough surface is the sum of the areas of several microcontact asperities, so the relatively high microcontact pairs will be the first to enter the plasticity. In the micrometer scale plastic behavior, the contact plastic behavior will reflect the size effect. The role of size-dependent parameters in the contact and friction properties of rough surfaces is a focus of future research.

(2) The study of interface contact and friction has a trend from static to dynamic. From the viewpoints of simulation and experimental methods, It is of great significance to reveal the cause of initiation of interface sliding, the reason of occurrence of interface sliding, and the factors affecting the sliding, so as to promote the research of friction mechanics.

(3) The research of interface contact and friction has a trend from qualitative to quantitative. Starting from the classical friction law, researchers not only reveal the causes and influencing factors of friction qualitatively, but also devote themselves to putting forward prediction formulas and criteria that can be directly used in engineering practice. The quantitative prediction of interfacial friction is of great significance for extending the study of contact mechanics and friction mechanics to engineering practice.

In order to make it more clearly, we have added some explanations in Section 5 in the revised manuscript. The revisions have been highlighted in blue.

We thank the reviewer again for your suggestions.

Reviewer 3 Report

Thank you for your efforts in compiling this handy review of the literature. It will give courage to new researchers to carry out further investigations and modelling. I found a few errors in the referencing. Please correct this.

line 161, 189, 259, 281, 485, 563: Error link/Missing links to citation numbers
line 240: Bush et al [29] =/= Zisis

Author Response

Thank you for your efforts in compiling this handy review of the literature. It will give courage to new researchers to carry out further investigations and modelling. I found a few errors in the referencing. Please correct this.

Point 1: line 161, 189, 259, 281, 485, 563: Error link/Missing links to citation numbers.

Response 1: We thank the reviewer for this comment. We have checked and revised the corresponding citations in the manuscript. The text has been highlighted in blue.

Point 2: line 240: Bush et al [29] =/= Zisis.

Response 2: We thank the reviewer for this comment. We have checked and revised the contents in the manuscript. The text has been highlighted in blue.

Round 2

Reviewer 1 Report

For me, the annotations of the first review were not sufficiently taken into account. I want to give you some examples:

(A) Regarding the drop test of mobile phones: In my eyes, this is not about friction but about contact mechanics. No coefficient of friction is mentioned in this paper. Even, there is no investigation on real rough surfaces described in paper [6]. In the source you mentioned smooth rigid surfaces are assumed - so the context for this paper is not given. This shows the problem: Contact and friction should not be mixed up although they are connected very closely.

(B) The sources of the images in figure 1 have to be mentioned whether they are cited form an online source or a scientific paper. In my opinion, the figure should be omitted, as it is only for illustration (as justified by the authors). This is not scientific standard and is also critical regarding Copyright. Three out of four pictures are not even mentioned in the text.

(C) Figure 13: Pictures are not cited (see (B)).

(D) I can't see any new References out of the last five years as you promised. For a Review Paper there should be more sources out of the last years. You promise to give an overview of the latest developments in the abstract.

(E) The style (bold/not bold) of the captions is still not consistent.

One new point out of your new Prospects:

(F) Point (2) of your Prospects: The research of the friction behavior of rough surfaces is no current or future trend. It is subject of research for many machine elements for years - static and dynamic friction.

Due to the lack of current sources and the and inadequate consideration of the annotations (see examples above), I recommend a major revision of the paper. The purpose of this paper is still not clear after the first revision.

Author Response

Response to Reviewer 1 Comments

For me, the annotations of the first review were not sufficiently taken into account. I want to give you some examples:

(A) Regarding the drop test of mobile phones: In my eyes, this is not about friction but about contact mechanics. No coefficient of friction is mentioned in this paper. Even, there is no investigation on real rough surfaces described in paper [6]. In the source you mentioned smooth rigid surfaces are assumed - so the context for this paper is not given. This shows the problem: Contact and friction should not be mixed up although they are connected very closely.

Response: We thank the reviewer for this comment.

We fully agree with the reviewer about this case of application. This paper focuses on the tribological research based on contact mechanics, but we apologize for not stating clearly and causing confusion to the readers. The case of mobile phone drop test is related to the mechanism of contact mechanics. The original purpose of this example is to express the wide application of surface mechanics represented by contact and friction in engineering, but it seems to have caused confusion. So we got rid of that example. The revised text has been highlighted in red in the manuscript.

We thanks the reviewer again for your patience and allowing us to consider this in depth.

(B) The sources of the images in figure 1 have to be mentioned whether they are cited form an online source or a scientific paper. In my opinion, the figure should be omitted, as it is only for illustration (as justified by the authors). This is not scientific standard and is also critical regarding Copyright. Three out of four pictures are not even mentioned in the text.

Response: We thank the reviewer for this comment.

We fully agree with the reviewer. We removed Figure 1 to avoid unnecessary copyright issues, and we apologize for this carelessness.

(C) Figure 13: Pictures are not cited (see (B)).

Response: We made this picture by ourselves to illustrate the prospects of friction research. This has already been mentioned in our former reply. Therefore, there is no need to cite the references.

(D) I can't see any new References out of the last five years as you promised. For a Review Paper there should be more sources out of the last years. You promise to give an overview of the latest developments in the abstract.

Response: We thank the reviewer for this comment. Since we revised the title and the purpose of this article, we did not make big revisions in the references. In view of your comments, we have revised the reference list. Of all the citations, 25% were in the last five years, and 50% were key researches to advance the development of this field in the last ten years. The revised text has been highlighted in red in the manuscript.

(E) The style (bold/not bold) of the captions is still not consistent.

Response: We thank the reviewer for this comment. We have revised and checked this format problem. The revised text has been highlighted in red in the manuscript.

One new point out of your new Prospects:

(F) Point (2) of your Prospects: The research of the friction behavior of rough surfaces is no current or future trend. It is subject of research for many machine elements for years - static and dynamic friction.

Response: We thank the reviewer for this comment.

We fully agree with the reviewer. In the prospects section, we removed the relevant statement that friction development trend has a static to dynamic trend. In the revised text, we state that tribology has a tendency from a single mechanical discipline to highly interdiscipline, and we gave some examples of tribology and other disciplines to form new disciplines. The revised text has been highlighted in red in the manuscript.

Due to the lack of current sources and the and inadequate consideration of the annotations (see examples above), I recommend a major revision of the paper. The purpose of this paper is still not clear after the first revision.

Round 3

Reviewer 1 Report

1) In my opinion, the newly chosen wording of the title does not sound correct. I suggest to change "standing on" to a more familiar wording (for example: on the basis of?) - I suggest proof reading of a native speaker.

2) In the title "Surfaces" should be written with a Capital letter according to the other words in the title.

3) Regarding Figure 12: If all those small images within your new graphic were made on your own and are not used in other publications before, it is okay - if not, please add the sources.

4) The heading "6. Conclusions" should be on the same page as the following text. I suggest a page break.

After these revisions, I

Author Response

1) In my opinion, the newly chosen wording of the title does not sound correct. I suggest to change "standing on" to a more familiar wording (for example: on the basis of?) - I suggest proof reading of a native speaker.

Response: We thank the reviewer for this comment.

We have revised the title. In addition, we have read the whole text and revised some of the expressions. The revised text has been highlighted in green.

2) In the title "Surfaces" should be written with a Capital letter according to the other words in the title.

Response: We thank the reviewer for this comment. We have revised the title.

3) Regarding Figure 12: If all those small images within your new graphic were made on your own and are not used in other publications before, it is okay - if not, please add the sources..

Response: We thank the reviewer for this comment.

One of the pictures (schematic of AFM method) is made by Yanwei Liu (our corresponding author). Other pictures have been cited from references (No. 158-162). The revised text has been highlighted in green.

4) The heading "6. Conclusions" should be on the same page as the following text. I suggest a page break.

Response: We thank the reviewer for this comment.

We have revised the format and the heading has been on the same page as the following text.
